# Therapeutic Treatment Options for In-Transit Metastases from Melanoma

**DOI:** 10.3390/cancers16173065

**Published:** 2024-09-03

**Authors:** Francesco Russano, Marco Rastrelli, Luigi Dall’Olmo, Paolo Del Fiore, Carlomaria Gianesini, Antonella Vecchiato, Marcodomenico Mazza, Saveria Tropea, Simone Mocellin

**Affiliations:** 1Soft-Tissue, Peritoneum and Melanoma Surgical Oncology Unit, Veneto Institute of Oncology IOV-IRCCS, 35128 Padova, Italy; marco.rastrelli@unipd.it (M.R.); luigi.dallolmo@unipd.it (L.D.); paolo.delfiore@iov.veneto.it (P.D.F.); antonella.vecchiato@iov.veneto.it (A.V.); marcodomenico.mazza@iov.veneto.it (M.M.); saveria.tropea@iov.veneto.it (S.T.); 2Department of Surgical, Oncological and Gastroenterological Sciences (DISCOG), University of Padua, 35128 Padova, Italy; carlomaria.gianesini@iov.veneto.it

**Keywords:** in-transit metastases, melanoma, surgery, immune checkpoint inhibitors, Isolated Limb Perfusion, electrochemotherapy

## Abstract

**Simple Summary:**

In-transit metastases (ITM) are a challenging aspect of advanced melanoma, traditionally treated with surgery. However, recent advances in systemic therapies, such as immune checkpoint inhibitors and targeted treatments, have significantly improved patient outcomes. These modern therapies are now often preferred over surgery alone. This article reviews the benefits of combining systemic and locoregional treatments, highlighting their potential to enhance survival and quality of life for patients with ITM. By integrating these approaches, we aim to provide a comprehensive strategy for optimizing melanoma treatment outcomes.

**Abstract:**

In-transit metastases (ITM) in melanoma present a significant therapeutic challenge due to their advanced stage and complex clinical nature. From traditional management with surgical resection, ITM treatment has evolved with the advent of systemic therapies such as immune checkpoint inhibitors and targeted therapies, which have markedly improved survival outcomes. This study aims to review and highlight the efficacy of both systemic and locoregional treatment approaches for ITM. Methods include a comprehensive review of clinical studies examining the impact of treatments like immune checkpoint inhibitors, targeted therapies, Isolated Limb Perfusion, and electrochemotherapy. The results indicate that combining systemic therapies with locoregional treatments enhances both local disease control and overall survival rates. The introduction of modern immunotherapies has not diminished the effectiveness of locoregional therapies but rather improved patient outcomes when used in conjunction. The conclusions emphasize that a multidisciplinary approach integrating systemic and locoregional therapies offers a promising strategy for optimizing the management of ITM in melanoma patients. This integrated treatment model not only improves survival rates but also enhances the quality of life for patients, suggesting a shift in standard care practices toward more comprehensive therapeutic regimens.

## 1. Introduction

In-transit melanoma metastases represent a distinct pattern of regional metastasis that occurs when melanoma cells spread through the lymphatic system from the primary tumor site to areas between the primary lesion and the regional lymph nodes. Unlike distant metastases, which spread through the bloodstream to other organs, ITM is confined to the lymphatic channels. These metastases are typically located more than 2 cm away from the primary tumor but do not reach beyond the nearest regional lymph node basin. ITMs represent a significant therapeutic challenge in advanced melanoma. Recent advancements in systemic therapies, including immune checkpoint inhibitors and targeted treatments, have notably improved patient outcomes. This review comprehensively explores the efficacy of both systemic and locoregional treatments for ITM, highlighting the synergistic potential of combining these approaches to enhance survival and quality of life. By integrating modern immunotherapies and locoregional therapies, the article advocates for a multidisciplinary strategy to optimize the management of ITM in melanoma patients, proposing a shift toward more comprehensive therapeutic regimens.

## 2. Definition and Clinical Presentation

ITM refers to a specific pattern of metastatic spread in cancer, particularly seen in melanoma and other skin cancers. In the 8th edition of the American Joint Committee on Cancer (AJCC) staging system for cutaneous melanoma, in-transit metastases (ITMs) are described as metastases in the skin or subcutaneous tissue that appear more than 2 cm from the original melanoma, situated between the primary tumor and the regional lymph nodes [1,2]. Melanoma typically spreads through the lymphatic system, and this progression can be mapped by tracing the lymphatic pathways from the primary tumor to the nearby lymph nodes. To assess the risk of regional metastasis, a sentinel lymph node biopsy is often performed to locate the first lymph node that the tumor drains into, providing key information on the potential spread of the disease [3]. Approximately 4% of patients with invasive cutaneous melanoma develop ITM, with the likelihood increasing to 11% in those with thicker primary tumors [1,4,5]. The AJCC Melanoma Staging System classifies ITMs into stage IIIB, IIIC, or IIID, depending on the extent of lymph node involvement, the thickness of the primary tumor, and the presence of ulceration. Specifically, stages IIIB, IIIC, and IIID reflect increasing severity and poorer prognosis based on these factors. Stage IIIB involves fewer lymph node involvements and may or may not have ulceration, while stages IIIC and IIID represent progressively more extensive lymph node involvement and more severe ulceration status [2]. ITMs can present in an extremely heterogeneous manner, varying in number, pigmentation (flesh-colored, pigmented, or erythematous), size, and distance from the primary site. They may appear as multiple cutaneous and subcutaneous nodules or as blisters, making diagnosis and clinical management challenging. They are predominantly found on the lower limbs—identifiable as isolated lesions or in clusters—but can occur on any part of the body depending on the primary tumor location and lymphatic drainage patterns. The nodules can exhibit various growth patterns, including rapid enlargement or a more indolent course. While some patients may be asymptomatic, others may experience symptoms such as pain, itching, or ulceration at the site of the nodules. Ulcerated lesions can lead to secondary infections and may require additional medical management. A microsatellite metastasis is a focus of metastatic tumor in the dermis or subcutis that is adjacent to but discontinuous with the primary melanoma and is identified during histopathologic assessment of the primary tumor excision. A satellite metastasis is a clinically evident cutaneous or subcutaneous metastasis that is within 2 cm of, but discontinuous with, the primary tumor. An in-transit metastasis is a clinically evident cutaneous or subcutaneous metastasis greater than 2 cm from the primary melanoma and typically situated between the primary melanoma and the regional lymph node basin [6,7,8]. 

ITMs are considered the result of inherently adverse tumor biology. Several factors are associated with an increased risk of developing ITMs, including the thickness of the primary tumor, ulceration, primary tumor location on the lower limbs, and advanced age of the patient. It has been observed that patients with ITMs tend to have a worse prognosis, with 5-year survival rates ranging from 52% to 81% and 10-year survival rates from 43% to 75%. In-transit metastases are associated with an unfavorable prognosis. The overall survival of patients with ITMs varies significantly based on the staging of the melanoma and the response to treatment. Patients with stage N3c disease have the worst outcomes, with lower survival rates. In-transit metastases pose a significant challenge in the management of advanced cutaneous melanoma [8]. Their clinical and biological complexity requires an integrated and personalized therapeutic approach to optimize both the quality of life and the survival of patients. Diagnosing ITM involves a combination of clinical examination and imaging. Dermatologists and oncologists may use dermoscopy to evaluate the surface characteristics of the lesions. Advanced imaging techniques, such as ultrasound, Positron Emission Tomography/Computed Tomography (PET-CT), and Magnetic Resonance Imaging (MRI), are employed to assess the extent of disease, detect deeper tissue involvement, and rule out distant metastases.

## 3. Molecular and Clinical ITM Aspects

ITM in melanoma is a unique and challenging subset of metastatic disease characterized by its distinct molecular and clinical features. Nakayama et al. provided early evidence of clonal origins of ITM, indicating specific genetic alterations unique to this form of metastasis, which really underscores the importance of understanding these molecular mechanisms for effective treatments [9]. Along the same line, strong evidence support peculiar molecular genetics and therapeutic resistance in melanoma [10], paving the way for the development of specialized treatment strategies in ITM. Notably, you may use clinicopathological characteristics to predict recurrence and survival in patients with in-transit metastases, thus suggesting that key features specific to ITM do exist [11]. 

ITM frequently harbors mutations in the NRAS gene, especially the Q61 mutation, which is significantly more common in ITM compared to other metastatic sites such as distant metastases [12,13]. Interestingly, antibodies with some selectivity for Q61R forms of NRAS have been developed that may have diagnostic purposes [14]. This mutation plays a crucial role in activating the MAPK pathway, which drives aggressive tumor growth and locoregional spread [15,16]. Intriguingly, gene analysis showed a lower incidence of NF-1 mutations in ITM relative to metastatic samples [13]. Additionally, ITM is often associated with a lower tumor mutational burden (TMB), which has been linked to a higher propensity for locoregional recurrences, including ITM, rather than distant metastasis. The combination of low TMB and NRAS mutations suggests that ITM tumors may possess unique molecular mechanisms [17]. From a biochemical perspective, ITM tumors also exhibit unique lipidomic profiles, with an increase in shorter-chain GM3 gangliosides compared to distant metastases. This alteration in lipid composition may enhance the tumor cells’ ability to survive and proliferate within the lymphatic environment, potentially serving as a biomarker or therapeutic target [18]. The immune microenvironment in ITM is notably distinct as well, with heterogeneous immune cell infiltration, including macrophages and dendritic cells, which may contribute to the tumor’s ability to evade the immune system and persist in the host [12]. Additionally, gene expression studies have revealed that ITM is marked by the upregulation of interleukin-8 (IL8), a cytokine known to promote angiogenesis and suppress immune responses. This upregulation further facilitates the growth and survival of ITM, making IL8 a potential target for therapeutic intervention [19]. Clinically, ITM presents a significant challenge due to its aggressive nature and its tendency to recur within the regional lymphatic system. The distinct molecular profile of ITM, including its NRAS mutations, specific lipid compositions, and unique immune environment, suggests that targeted therapies addressing these specific pathways could be more effective than traditional treatments. For example, therapies that inhibit the MAPK pathway or target IL8 may offer new avenues for managing ITM. Moreover, the role of TMB in predicting recurrence emphasizes the importance of integrating molecular profiling into the clinical management of ITM. By regularly assessing mutational burden and immune signatures, clinicians can better tailor treatment strategies to the individual patient, potentially improving outcomes by addressing the specific biological mechanisms driving ITM. Despite these advances, ITM remains a highly complex and understudied form of melanoma. Continued research is essential to fully understand the molecular underpinnings of ITM and to develop novel therapeutic strategies that can effectively combat this aggressive cancer. 

## 4. Systemic Treatments for In-Transit Metastases from Melanoma

### 4.1. Immune Checkpoint Inhibitors

Immune checkpoints are molecules on T cells that regulate the immune response, preventing autoimmunity and reducing tissue damage. Among the most studied checkpoints is Cytotoxic T-Lymphocyte-Associated Protein 4 (CTLA-4), which manages early T-cell activation by competing with CD28, and Programmed Death-1 (PD-1), which regulates T-cell activity in peripheral tissues, particularly during cancer progression. Immune checkpoint inhibitors, such as ipilimumab (a CTLA-4 inhibitor) and pembrolizumab/nivolumab (PD-1 inhibitors), interfere with these checkpoints, enhancing T-cell capacity to destroy cancer cells. These inhibitors have revolutionized cancer treatment, especially for difficult-to-treat tumors (Figure 1) [20].

#### 4.1.1. Ipilimumab (Anti-CTLA-4)

Ipilimumab [21,22,23] is a fully human monoclonal antibody that targets CTLA-4, enhancing T-cell activation (Figure 1) and proliferation, which helps the immune system better identify and attack cancer cells [24,25]. Approved by the U.S. Food and Drug Administration for unresectable or metastatic melanoma in early 2011, it soon received European Commission approval [26]. This marked a shift from dacarbazine, the long-standing standard treatment [27]. An elegant study demonstrated that adding ipilimumab to dacarbazine improves overall survival (OS) in patients with metastatic melanoma [22]. For instance, a multicenter study reported a 30% complete response and 43% overall response rate in patients treated with ipilimumab, with a median progression-free survival of nine months and melanoma-specific survival rates of 83% at one year [28]. Another study showed a 38% overall response rate with ipilimumab alone in ITM patients [29]. Despite its efficacy, Ipilimumab can cause a range of Immune-Related Adverse Events (irAEs), such as dermatitis, colitis, hepatitis, and endocrinopathies [30]. In the most severe cases, Ipilimumab has a fatality rate of approximately 1.08% in melanoma patients, primarily due to colitis/diarrhea, which accounts for 70% of these deaths [31]. Notably, recent research suggested that the occurrence of irAEs during ipilimumab monotherapy among melanoma patients may serve as a marker of an improved anti-tumor response [32].

#### 4.1.2. Nivolumab and Pembrolizumab (Anti PD-1)

The next phase of immune checkpoint inhibitor approvals focused on anti-PD-1 drugs, beginning with pembrolizumab in 2014 for metastatic melanoma, followed by nivolumab in 2015. Nivolumab is a fully human IgG4 monoclonal antibody, while pembrolizumab is a humanized IgG4 monoclonal antibody. Both block the PD-1 receptor, preventing interaction with its ligands PD-L1 and PD-L2, thereby keeping T cells active against tumors [33,34]. In advanced melanoma, nivolumab showed superior 1-year survival (72.9% vs. 42.1%) and progression-free survival (5.1 vs. 2.2 months) over dacarbazine [35]. Pembrolizumab outperformed ipilimumab with better survival rates and fewer severe side effects (13.3% vs. 19.9%) [36]. In ITM patients, pembrolizumab and nivolumab achieved a 36% complete response rate and 10-month progression-free survival, indicating greater effectiveness than anti-CTLA-4 treatments [28]. Eventually, both nivolumab and pembrolizumab frequently cause a range of common side effects such as fatigue, rash, diarrhea, nausea, and musculoskeletal pain. While generally manageable, these symptoms can significantly impact patients’ quality of life [37].

#### 4.1.3. Combination of Immune Checkpoint Inhibitors 

Studies have shown that combining PD-1 and CTLA-4 inhibitors improves treatment outcomes for patients with in-transit melanoma. The CheckMate trials (069, 067, and 511) demonstrated that dual therapy with nivolumab and ipilimumab significantly enhances overall response rates and progression-free survival compared to monotherapy, albeit with a higher incidence of adverse events [38,39,40,41,42]. Specifically, in the context of ITM, ipilimumab, combined with nivolumab, was used as a neoadjuvant therapy for the treatment of four patients with in-transit melanoma. The combination of drugs, administered before surgery, showed promising results in reducing tumor burden and achieving pathological responses in patients. The study indicated that this regimen is effective and well-tolerated, with all treated patients achieving disease control and no relapses during follow-up [43]. Similarly, another study on 287 ITM patients demonstrated that the combination of immune checkpoint inhibitors is highly effective. The overall response rate was highest with combination therapy at 68%, compared to 56% for PD-1 inhibitors and 43% for CTLA-4 inhibitors [28].

### 4.2. Targeted Therapies

Targeted therapies refer to treatments specifically designed to target molecular abnormalities or specific pathways that are essential for the growth and survival of melanoma cells. Unlike traditional chemotherapy, which affects all rapidly dividing cells, targeted therapies aim at particular genes, proteins, or the tissue environment that contributes to cancer development and progression. As a matter of fact, targeted therapies have changed the treatment paradigm for patients with BRAF V600 mutations, significantly improving clinical outcomes. 

#### BRAF and MEK Inhibitors

BRAF and MEK inhibitors are targeted therapies primarily used to treat cancers with specific genetic mutations, notably in melanoma. BRAF inhibitors work by targeting and inhibiting the activity of the BRAF protein, specifically the mutant form BRAF V600E, which is commonly found in various cancers, including melanoma. This protein is part of the MAPK/ERK signaling pathway that controls cell division and differentiation. Mutations in BRAF lead to uncontrolled cell growth, which can result in cancer. BRAF inhibitors, such as Vemurafenib [44] and Dabrafenib [45], block the mutant BRAF protein, reducing uncontrolled cell growth [46]. The side effects often associated with BRAF inhibitors include skin rashes, photosensitivity, joint pain, fatigue, nausea, and secondary skin cancers like squamous cell carcinoma [47]. MEK inhibitors target downstream enzymes in the MAPK/ERK pathway, further inhibiting cancer cell proliferation [48]. Common MEK inhibitors include Trametinib [49] and Cobimetinib [50]. Typical side effects of MEK inhibitors are rash, diarrhea, peripheral edema, fatigue, nausea, and cardiomyopathy [49,50].

In in-transit melanoma with BRAF V600 mutations, combining BRAF and MEK inhibitors, such as Dabrafenib and Trametinib, has shown significant therapeutic benefits. A five-year study reported a 34% overall survival and 19% progression-free survival, with higher survival rates in patients with a complete response [51]. The combination of Vemurafenib and Cobimetinib in treating BRAF V600 mutation-positive metastatic melanoma improves outcomes significantly compared to Vemurafenib alone. A phase 3 trial showed a median progression-free survival of 9.9 months and an overall response rate of 68% with the combination, compared to 6.2 months and 45% with Vemurafenib alone. While the combination therapy increased side effects like diarrhea and photosensitivity, it also reduced the risk of secondary skin cancers [50].

### 4.3. Chemotherapy

Chemotherapy is a type of cancer treatment that uses drugs to destroy cancer cells. It works by targeting rapidly dividing cells, a characteristic of cancer cells, and interrupting their growth and reproduction. However, its action is nonspecific (both tumor and healthy cells may be targeted), resulting in significant side effects such as bone marrow suppression, gastrointestinal issues, and hair loss. Compared to targeted therapies and immunotherapies, chemotherapy is less effective due to the cancer’s ability to develop drug resistance, genetic heterogeneity within tumors, and robust DNA repair mechanisms. Nonetheless, chemotherapy remains a component of the treatment for advanced melanoma, although it is generally reserved for patients who do not respond to other therapies or who have contraindications to these treatments. Chemotherapy can be administered orally, intravenously, or directly into the affected area, depending on the type and stage of cancer [52].

#### 4.3.1. Dacarbazine 

Dacarbazine (DTIC) is an alkylating agent used in chemotherapy primarily for treating metastatic melanoma and Hodgkin’s lymphoma. Dacarbazine works by alkylating DNA, which disrupts DNA replication and leads to cell death. It was approved by the FDA in the 1970s and remains one of the few chemotherapy agents approved for melanoma. Despite its long-standing use, DTIC effectiveness in advanced melanoma is limited, with response rates of 10–20% and a median overall survival of 9–11 months. Common side effects include nausea, vomiting, myelosuppression, and fatigue, which can significantly impact patient quality of life. In recent years, the use of Dacarbazine has declined with the advent of more effective treatments like targeted therapies and immunotherapies. However, Dacarbazine may still be used in certain cases [53,54], such as when newer treatments are not available, not suitable, or as part of combination therapy approaches to enhance treatment efficacy [52,55].

#### 4.3.2. Temozolomide and Fotemustine

Fotemustine is a nitrosourea alkylating agent that can cross the blood–brain barrier, making it potentially useful for treating melanoma metastases in the brain. It works by forming cross-links in DNA, thus preventing DNA replication and leading to cell death [56]. Fotemustine has shown some efficacy in advanced melanoma, particularly in patients with brain metastases, but its overall effectiveness is limited. Side effects include myelosuppression, liver enzyme abnormalities, and gastrointestinal symptoms. Despite its ability to cross the blood–brain barrier, its overall impact on survival and disease progression in melanoma patients is relatively modest [57,58].

Temozolomide is an oral alkylating agent that works by methylating DNA, which leads to DNA damage and subsequent cancer cell death. It is structurally similar to dacarbazine but can be administered orally, offering more convenience, and with the added benefit of penetration into the central nervous system. Despite its convenience, the efficacy of Temozolomide in advanced melanoma is modest. Accordingly, clinical studies have shown that it can produce some responses in metastatic melanoma, but the overall survival benefits are generally low. Its side effects include nausea, vomiting, fatigue, and myelosuppression, which limit the dose that can be safely administered [59,60]. 

These agents have shown some effectiveness in treating advanced melanoma, with variable response rates. They are considered when managing melanoma with brain metastases or when patients cannot tolerate newer targeted therapies or immunotherapies. They are often used in patients who do not tolerate other drugs [52,61].

## 5. Locoregional Treatments

Locoregional treatments for in-transit melanoma are therapeutic strategies designed to control melanoma that has spread locally to the skin or nearby lymph nodes. These treatments include surgical excision to remove tumors, Isolated Limb Perfusion and Infusion, which deliver high doses of chemotherapy directly to affected limbs, electrochemotherapy to target and destroy cancer cells in specific regions, and intralesional therapies where drugs are injected directly into the melanoma lesions. The primary objective of these treatments is to manage and limit the progression of melanoma within a localized area, improve patient outcomes, and reduce the likelihood of recurrence. Of note, these treatments may also serve a palliative purpose, aiming not to cure but to alleviate symptoms and maintain the patient’s quality of life by controlling the growth of local tumors. This approach helps to prevent the development of disfiguring or symptomatic lesions, thereby improving the patient’s comfort and appearance. Locoregional treatments are often used in conjunction with systemic therapies to provide comprehensive management of the disease.

### 5.1. Isolated Limb Perfusion (ILP)

Isolated Limb Perfusion (ILP) or Hyperthermic Isolated Limb Perfusion (HILP) is a locoregional treatment used to achieve limb-sparing disease control in advanced melanoma and in-transit metastases of the limbs. The procedure, performed under general anesthesia, involves isolating the blood circulation of the affected limb from the rest of the body, allowing high doses of chemotherapeutic agents to be delivered directly to the tumor site without causing systemic toxicity. It is a treatment that involves vessel isolation of the limb to treat (iliac or femoral in the lower limb and axillary for the upper limb) and their cannulation to connect the limb to an extracorporeal circulation, in order to obtain a closed circuit. A tourniquet and vascular clamps prevent blood and drugs (introduced into the closed circuit) from reaching the systemic circulation with consequent toxicity. To maximize the efficacy of the treatment, drugs (typically melphalan, sometimes in combination with tumor necrosis factor-alpha) are used at high doses, which cannot be achieved by systemic therapy, and the limb is heated to a controlled temperature (via sensors) of 38–41 °C [25,62,63,64]. The elevated temperature helps to increase the permeability of tumor cells to the drug, potentially leading to more effective treatment outcomes [65]. ILP is particularly useful for bulky, multiple, or recurrent in-transit melanoma metastases that cannot be managed with surgical excision. Accordingly, clinical studies have shown that ILP can be a safe and effective therapy for treating ITM from melanoma in patients with non-resectable disease. In a systematic review of 22 studies including 2018 procedures, ILP showed a complete response rate of 58%, an ORR of 90%, and a 5-year OS of 37% [64]. The efficacy of ILP was further confirmed by a study from Rossi et al., which found a higher CR in patients treated with a combination of TNF-alpha and Melphalan compared to those treated with Melphalan alone (60.3% vs. 41.5%) [62]. However, when it comes to treating melanoma metastases in the pelvic region, the two techniques used worldwide, such as Isolated Limb Perfusion and Isolated Limb Infusion (see below), cannot effectively target metastases. Pelvis relapse occurs in 15% of metastatic cutaneous melanoma and in this case, a locoregional treatment option is hypoxic pelvic perfusions with hemofiltration [66,67,68]. Despite this limitation, ILP remains a highly effective technique for targeting and controlling melanoma metastases in limbs.

### 5.2. Isolated Limb Infusion (ILI)

ILI is a regional chemotherapy technique used to treat advanced or in-transit melanoma confined to a limb. The procedure uses catheters inserted percutaneously into the limb’s main artery and vein, allowing for a high concentration of the drug to be administered locally. A tourniquet is used to reduce the risk of leakage of drugs in the systemic circulation. ILI is less invasive than ILP and does not require an open surgical approach or extracorporeal circulation. This makes it suitable for patients who may not tolerate more invasive procedures, such as those who are elderly or have multiple comorbidities. Thus, this procedure avoids the insurgence of the most common complications associated with a major surgery and can be more easily performed in patients with complex clinical conditions. The Isolated Limb Infusion approach should indeed be considered a valid option for a wide range of patients as an international study on elderly patients demonstrated that ILI is safe and effective, with similar response rates between older and younger patients [69]. The chemotherapy drugs commonly used in ILI include melphalan and, occasionally, actinomycin D. It is particularly useful for patients with unresectable, recurrent, or extensive in-transit melanoma metastases. While ILI can achieve significant tumor responses, it is typically performed in specialized centers due to the expertise and equipment required [25,70,71]. The international multicenter study on ILI for stage 3B and 3C melanoma, involving 687 patients, showed promising results. The overall response rate was 64.1%, with a complete response observed in 28.9% of patients. Significant limb toxicity occurred in 3.9% of cases, but no amputations were necessary. The median in-field progression-free survival was 10.1 months, the median distant progression-free survival was 28.6 months, and the median overall survival was 38.2 months. Patients who achieved a complete or partial response had significantly better outcomes, with longer median survival compared to non-responders [72]. These findings strongly endorse the use of ILI for treating advanced melanoma.

### 5.3. Electrochemotherapy (ECT)

ECT is an effective treatment modality for in-transit melanoma, combining chemotherapy and electric pulses to enhance drug uptake by cancer cells. It is used for cutaneous and subcutaneous metastases, where it increases the permeability of cell membranes, allowing higher concentrations of chemotherapeutic agents like bleomycin or cisplatin to enter the cells. This localized treatment results in high response rates with minimal systemic toxicity, making it suitable for patients with multiple small metastases or larger resistant tumors. Studies have shown that ECT can achieve significant tumor regression and symptom relief, providing a valuable option for managing advanced melanoma. ECT combines the injection of chemotherapeutic drugs, such as bleomycin or cisplatin, with the application of electrical pulses. The European Standard Operating Procedures for Electrochemotherapy (ESOPE) indicate that this technique achieves an objective response in about 80% of cases, with complete tumor responses in 60–70% of treatments. Indeed, ECT has shown significant tumor reduction, symptom relief, and improved quality of life, making it a valuable option for advanced melanoma, particularly when other therapies have failed [73,74]. Importantly, a 2021 systematic review and meta-analysis involving 1161 patients with metastatic cutaneous melanoma confirmed ECT effectiveness, with one-year local control rates between 54% and 89% and overall survival rates from 67% to 89%. While ECT mainly causes local side effects like pain and skin toxicity (erythema, edema, and ulceration), systemic toxicity is low, though severe cases like fatal respiratory failure may occur in those with pre-existing conditions [75]. In summary, ECT provides robust tumor control with a favorable safety profile for cutaneous melanoma metastases.

### 5.4. Intralesional Therapies

Intralesional therapies for in-transit melanoma involve the direct injection of therapeutic agents into melanoma lesions that are not suitable for surgical removal. These therapies include agents such as Bacillus Calmette-Guérin (BCG), interleukin-2 (IL-2), Granulocyte–Macrophage Colony-Stimulating Factor (GM-CSF), and talimogene laherparepvec (T-VEC). These therapies offer a localized treatment approach, minimizing systemic exposure and reducing side effects while effectively targeting and destroying tumor cells within the treated area, thus providing local control of the tumor. They are particularly beneficial for patients with limited metastatic disease or those who are not candidates for systemic therapy. 

#### 5.4.1. Bacillus Calmette-Guérin

BCG is an attenuated strain of *Mycobacterium bovis* initially developed as a vaccine against tuberculosis. It was first recognized for its anti-tumor properties in the early 20th century when researchers observed a lower frequency of cancer in patients with tuberculosis. BCG has since been used as an immunotherapy for various cancers, most notably bladder cancer, where it induces a strong local immune response that helps to shrink tumors and prevent recurrence. In the context of melanoma, BCG can alter the tumor microenvironment to favor anti-tumor T-cell responses. Studies have shown that intralesional injection of BCG in melanoma can lead to significant regression of malignant nodules and improved patient survival [76,77,78]. BCG works by stimulating the immune system, increasing the infiltration of T cells into the tumor, and enhancing the production of cytokines such as IFN-γ, which play a crucial role in anti-tumor immunity. However, its use in melanoma has significantly declined due to its side effects along with the development of more effective treatments, such as immune checkpoint inhibitors and targeted therapies. The current focus on BCG is mainly on bladder cancer, where it remains a standard treatment. In melanoma, while BCG showed some promise, the advent of newer therapies with higher efficacy and better response rates has overshadowed its use. As a result, BCG is not commonly used as an intralesional therapy for melanoma today, with modern treatments like pembrolizumab, nivolumab, and oncolytic virus therapy taking precedence due to their superior outcomes and response rates [79]. 

#### 5.4.2. Interleukin-2

IL-2 is a pleiotropic cytokine produced by T lymphocytes that enhances the maturation of regulatory T cells, promotes the differentiation of CD4^+^ T cells into T helper-1 and T helper-2 cells, and activates the expression of genes controlling the inflammatory process. IL-2 therapy for in-transit melanoma involves injecting IL-2 directly into the melanoma lesions. This method leverages the immune-stimulating properties of IL-2 to provoke an immune response specifically at the tumor site. Intra-lesional IL-2 has been shown to produce high response rates with manageable side effects. Studies have demonstrated a complete response in 50% of patients and 78% of individual lesions [80], with side effects primarily consisting of localized pain, swelling, and mild flu-like symptoms. The treatment protocol typically involves bi-weekly injections over several cycles. In practice, patients have reported maintaining their daily activities with minimal disruption, and in many cases, non-injected bystander lesions also responded to the treatment. Overall, intra-lesional IL-2 is considered effective for managing in-transit melanoma due to its high response rates and favorable safety profile. It is particularly useful as an early line of treatment for in-transit melanoma before considering more toxic systemic therapies. Examples from clinical practice show that patients often achieve significant tumor regression and symptom relief with this treatment approach [80,81,82].

#### 5.4.3. Granulocyte–Macrophage Colony-Stimulating Factor

GM-CSF, also known as sargramostim, is an immunotherapy used in the treatment of melanoma. GM-CSF works by stimulating the immune system, specifically enhancing the activity and proliferation of macrophages and dendritic cells, which are critical for tumor recognition and destruction [83]. In the context of in-transit melanoma, GM-CSF can be administered intralesionally, where it is injected directly into the melanoma lesions. This localized treatment can stimulate a strong immune response at the tumor site, leading to tumor regression. Clinical studies have shown that patients receiving GM-CSF therapy can experience significant tumor reduction and improved overall survival rates. For example, one study involving patients with stages II(T4), III, and IV melanoma administered GM-CSF subcutaneously in 28-day cycles for three years. The results indicated that prolonged GM-CSF therapy was well tolerated, with the most common side effects being mild injection site reactions and flu-like symptoms. Importantly, the study reported a five-year melanoma-specific survival rate of 60%, highlighting the potential benefits of GM-CSF as an adjuvant therapy in reducing melanoma recurrence and improving survival outcomes [84]. GM-CSF is often used in combination with other therapies to improve efficacy. For instance, GM-CSF combined with the CTLA-4 inhibitor ipilimumab has been shown to prolong overall survival and lower toxicity in patients with unresectable stage III or IV melanoma. This combination therapy demonstrated a significant increase in one-year overall survival rates compared to ipilimumab alone, without a notable improvement in progression-free survival [85]. The ability of GM-CSF to enhance the immune system response to melanoma cells makes it a valuable option for managing in-transit melanoma, particularly in patients at high risk of recurrence or those who have not responded to other treatments.

#### 5.4.4. Talimogene Laherparepvec

In the context of in-transit melanoma, T-VEC is a well-known oncolytic viral therapy. This therapy involves injecting a genetically engineered herpes simplex virus type 1 (HSV-1) directly into melanoma lesions. The virus selectively replicates within the tumor cells, leading to their destruction and triggering a systemic immune response against the cancer [86]. T-VEC generates GM-CSF, boosting dendritic cell activity and enhancing anti-tumor immunity. In clinical use, T-VEC is injected into melanoma lesions, starting with 1 × 10^6^ PFU/mL and increasing to 1 × 10^8^ PFU/mL in subsequent bi-weekly cycles until all lesions are gone. For instance, a patient with PD-1 inhibitor-resistant in-transit melanoma achieved complete remission after 11 T-VEC cycles and remained disease-free for three years. [87]. The OPTiM phase III trial found that T-VEC had a 26.4% overall response rate and a 10.8% complete response rate [88]. While most side effects are mild, such as injection site reactions and flu-like symptoms, severe adverse events are uncommon. Researchers are also exploring T-VEC in combination with other treatments to boost its efficacy. When combined with immune checkpoint inhibitors like ipilimumab, T-VEC’s ability to release tumor antigens and promote immune cell activity can be enhanced by the checkpoint inhibitors, which prevent the immune system from being suppressed. [89]. Although the combination of T-VEC with pembrolizumab did not significantly extend progression-free or overall survival compared to pembrolizumab alone, it did result in a higher objective response rate [90]. These results suggest that T-VEC, whether used alone or alongside other immunotherapies, may offer a promising treatment option for in-transit melanoma, particularly for tumors that are resistant to other therapies. Combining T-VEC with checkpoint inhibitors could enhance anti-tumor immunity and lead to better patient outcomes, especially in those unresponsive to other immunotherapies.

### 5.5. Laser Treatments 

Laser treatment for in-transit melanoma, particularly using CO_2_ lasers, involves the application of focused high-energy laser light to ablate (remove) the tumor tissue. This method is especially effective for cutaneous metastases where surgical options are impractical. CO_2_ laser ablation works by vaporizing the water content within cells, leading to the precise removal of cancerous tissue with minimal damage to surrounding healthy tissue [91]. Clinical studies have shown that a CO_2_ laser can be effective in treating cutaneous melanoma metastases, particularly in superficial and accessible lesions. Vrielink et al. reported that CO_2_ laser ablation can significantly reduce the size of metastatic lesions, with good tolerability and minimal side effects [92]. Interestingly, this approach provides significant palliative benefits, offering symptomatic relief and local disease control. For example, studies have demonstrated that CO_2_ laser ablation can achieve prolonged survival and local disease-free intervals for patients with recurrent melanoma. One study showed that 54.8% of patients treated with CO_2_ laser ablation survived for a median duration of 5.4 years, while another reported that 62.5% of patients remained disease-free one year post-treatment [93]. Thus, the use of a CO_2_ laser is particularly indicated in patients with localized disease who are not ideal candidates for surgery or other forms of locoregional treatment.

### 5.6. Radiotherapy (RT)

RT is a significant treatment for in-transit melanoma metastases, particularly when surgical options are unfeasible. RT provides effective local control by reducing tumor burden and alleviating symptoms such as pain and bleeding. Additionally, combining RT with systemic therapies, like ICIs, has shown promising results, enhancing both local and systemic disease control. Studies, including the RTOG 83-05 trial, have influenced the use of higher radiation doses per fraction for melanoma, improving locoregional control. When combined with ICIs, RT can induce immunogenic cell death, enhancing the immune response against melanoma cells. This synergistic effect has led to better outcomes, such as improved progression-free survival and overall response rates. Accordingly, a recent study analyzed the impact of RT on locoregional recurrence in 71 patients with stage III melanoma after adjuvant immunotherapy. It found that adjuvant RT significantly reduced the rate of locoregional recurrence (8% vs. 36%) and improved locoregional recurrence-free survival. Toxicity from RT was mostly mild, with no grade 3 or higher toxicity reported [94]. However, while adjuvant RT increases local control, it has not significantly impacted overall survival or distant metastasis-free survival when effective systemic therapies are available. Overall, the integration of RT with systemic treatments offers a comprehensive approach to treating ITM, maximizing therapeutic efficacy, and improving patient outcomes [95].

## 6. Current Guidelines and Recommendations for the Treatment of In-Transit Melanoma Metastases

The management of in-transit melanoma metastases involves a variety of approaches, as detailed in the current guidelines and recommendations from major oncology organizations.

### 6.1. National Comprehensive Cancer Network (NCCN) Guidelines

The NCCN Guidelines (Version 2.2024) provide a series of detailed recommendations for managing the treatment of stage III cutaneous melanoma with satellite and in-transit metastases. The primary recommendation is to perform complete surgical excision of satellite and in-transit metastases to achieve clear margins and minimize the risk of local recurrence, potentially followed by adjuvant systemic therapies. The decision between observation and adjuvant therapy should consider the risk of melanoma recurrence and the toxicity of the treatment.

#### 6.1.1. Adjuvant Therapy Options

Immune Checkpoint Inhibitors

Nivolumab: The CheckMate 238 study demonstrated a significant improvement in recurrence-free survival (RFS) compared to ipilimumab, with a better safety profile. Nivolumab is considered a category 2A option for patients with satellite and in-transit metastases that have been completely excised with clear margins [96,97];Pembrolizumab: Pembrolizumab is also recommended as an adjuvant treatment option post-excision. The KEYNOTE-054 study showed an improvement in RFS compared to placebo [98,99]. Similarly to nivolumab, pembrolizumab is recommended as a category 2A option.

#### 6.1.2. Targeted Therapies

Dabrafenib/Trametinib: Specifically for patients with BRAF V600 mutation. The COMBI-AD study showed an improvement in RFS for patients treated with the dabrafenib/trametinib combination compared to placebo [100,101,102]. This combination is recommended as a category 2A option for patients with satellite and in-transit metastases who have completed surgical excision. In the event of local recurrence, a new surgical excision is considered. Adjuvant options after complete excision again include nivolumab, pembrolizumab, and dabrafenib/trametinib, based on BRAF mutation and patient preference.

Other alternatives include systemic therapy or intralesional therapy, potentially followed by complete excision if possible. If radical surgical treatment is not possible or is performed without achieving clear margins, the NCCN guidelines preferentially recommend the use of systemic therapies such as immune checkpoint inhibitors and targeted therapies. Consideration of locoregional treatments (HILP or ILI), intralesional therapies (T-VEC or IL-2), or other local palliative treatments to alleviate symptoms is possible. Participation in clinical trials is strongly encouraged as the best management for patients with melanoma.

### 6.2. European Society for Medical Oncology (ESMO) Guidelines

The 2019 ESMO guidelines highlight the importance of a multidisciplinary approach for managing in-transit melanoma (ITM), combining locoregional and systemic treatments. While surgery remains an option for patients with resectable satellite or in-transit metastases, the increasing effectiveness of systemic therapies is shifting the focus, as surgery carries a risk of rapid disease progression that might undermine the long-term benefits of systemic treatments. Numerous clinical studies support the use of adjuvant systemic therapies, such as immune checkpoint inhibitors (anti-PD-1 or anti-CTLA-4) and BRAF/MEK inhibitors, particularly for patients with stage III melanoma. For unresectable satellite or in-transit metastases or inoperable primary limb tumors without additional metastases, Isolated Limb Perfusion with melphalan, with or without tumor necrosis factor-alpha (TNF-alpha), is recommended [Grade III, Recommendation C]. Additionally, T-VEC has demonstrated improved durable response rates compared to subcutaneous granulocyte–macrophage colony-stimulating factor, particularly in patients with stage IIIB/C and IVM1a melanoma (according to the seventh edition of the AJCC) [Grade I, Recommendation B] [103]. These local treatments should be carefully weighed against systemic options to ensure that the long-term benefits are not compromised. Given the need for significant surgical expertise or specialized knowledge in oncolytic virus use, such treatments should be confined to specialized centers. Radiotherapy, electrochemotherapy, carbon dioxide (CO_2_) laser, or other intralesional therapies, while less established, may also be considered within the context of clinical trials.

### 6.3. Associazione Italiana Di Oncologia Medica (AIOM) Guidelines 2023

The 2023 AIOM guidelines provide a structured and evidence-based approach for the treatment of these complex conditions. The diagnosis of in-transit metastases or satellite lesions is confirmed through cytological examination by fine needle aspiration biopsy (FNAB). Subsequently, it is crucial to restage the patient using appropriate imaging modalities, such as computed tomography or magnetic resonance imaging, to determine the extent of the disease and plan the optimal therapeutic strategy. Treatment options include surgery when lesions are resectable. The goal of the surgery is to completely excise the lesions with clear margins, in order to reduce the risk of local recurrence. However, surgery may be limited by the location and number of lesions. Locoregional therapies are indicated for non-resectable lesions or for patients in whom surgery is not feasible. The main locoregional therapies include the following:Electrochemotherapy: Used to treat inoperable lesions and skin metastases, even in locations other than the limbs;Radiotherapy: Can be employed to treat inoperable lesions, contributing to local disease control. Radiotherapy may be used alone or in combination with other locoregional and systemic therapies;HILP: Indicated for extensive limb lesions. This technique involves isolating the blood flow of the affected limb and infusing high doses of chemotherapy (melphalan) with/without TNF-alpha while the limb is heated. HILP is particularly effective for locally advanced and inoperable limb diseases.

Systemic therapies are crucial for managing in-transit metastases and satellite lesions, especially in the presence of advanced or non-resectable disease. Immune checkpoint inhibitors and targeted therapies have shown efficacy in improving recurrence-free survival and overall survival in patients with advanced melanoma. Systemic chemotherapy may be considered in selected cases, particularly when other therapeutic options are not feasible or as part of a combined approach. However, the use of chemotherapy is often limited due to its toxicity and variable response. Inclusion in clinical trials is strongly encouraged, as it provides patients access to new therapies and contributes to the advancement of scientific research.

## 7. Discussion

In-transit melanoma, metastases represent a complex therapeutic challenge and optimizing treatment requires an integrated and multidisciplinary approach. Current guidelines for the treatment of in-transit melanoma metastases involve both systemic and locoregional treatments. However, when the disease is not resectable with radical intent, the general recommendation is typically to commence systemic therapy. 

Extensive research has consistently demonstrated that immune checkpoint inhibitors [29] and BRAF/MEK inhibitors [100] significantly improve survival rates in patients with advanced melanoma, including stages III and IV (see above and Table 1). 

More in detail, Nan Tie et al. [29] conducted a study specifically focusing on in-transit melanoma metastases. This retrospective review encompassed 54 ITM patients treated with systemic immune checkpoint inhibitors, either as a monotherapy or in combination, across three Australian centers from 2013 to 2018. The study demonstrated durable responses, with overall survival and progression-free survival rates comparable to those observed in stage IV melanoma. Notably, the median progression-free survival was 11.7 months, affirming the effectiveness of ICIs for ITM and underscoring the potential for further investigation into their integration with other therapeutic approaches [29]. Similarly, the COMBI-AD trial investigated the effects of dabrafenib (a BRAF inhibitor) plus trametinib (a MEK inhibitor) on patients with resected BRAFV600-mutant stage III melanoma, including those with in-transit metastases. This prospective study included stage IIIA, IIIB, and IIIC patients and demonstrated that combination therapy significantly improved relapse-free survival compared to placebo, with a 4-year relapse-free survival rate of 54% versus 38% for placebo [100]. Overall, despite evidence that ITM patients respond well to these systemic treatments, the literature contains only a few ITM-focused studies and those that include ITM often lack detailed reporting on this subset. Consequently, there is limited organized data evaluating the efficacy of systemic treatments specifically for ITM patients.

It is important to highlight at this point that while ICIs and BRAF/MEK inhibitor therapies are highly effective systemically, achieving moderate local response rates of 30-40%, locoregional treatments can result in significantly higher response rates of 60-80% locally. This is particularly significant for ITM, as the disease is characterized by its spread within the regional lymphatic system and skin. Effective locoregional control is crucial for managing symptoms and improving patient outcomes.

Thus, despite the shift toward immune checkpoint inhibitors and systemic therapies as the preferred and recommended first-line options due to their efficacy and durable responses, locoregional treatments remain essential. In line with these considerations, many recent studies, both retrospective and prospective, are investigating the combined use of locoregional treatments and systemic drugs for ITM. Here, we report several studies that demonstrate not only the beneficial effects but also the potential synergistic effects of combining these treatments for ITM patients. Holmberg et al. analyzed the impact of ICIs in a larger cohort of 287 patients with in-transit melanoma metastases, with or without nodal involvement (AJCC8 N1c, N2c, and N3c), between 2015 and 2020. Their findings aligned with studies by Nan Tie et al., showing a similar progression-free survival of approximately 10 months. Notably, this study included patients treated with both ICIs and locoregional therapies; specifically, 98 out of 287 patients received radiotherapy, T-VEC, ILI, or ILP. Although the study could not definitively determine whether the combination of locoregional and systemic treatments provided a significant advantage over systemic treatment alone, it did indicate that the combination was not detrimental. Importantly, while progression disease (PD) rates were around 30% for patients who did not respond to ICIs, the addition of locoregional treatments such as ILP, ILI, T-VEC, and electrochemotherapy resulted in much lower PD rates (3–18%) in similar cases. The authors hypothesized that for patients with multiple, bulky, or rapidly recurrent ITM, a combination of locoregional and systemic treatments might be beneficial. Indeed, locoregional treatments, which do not negatively impact the efficacy of ICIs, may be particularly useful when ICIs alone are insufficient [28]. A recent study examined ITM patients who underwent locoregional ILP treatment between 2015 and 2020, comparing the outcomes between those who were ICI-naïve and those who had been treated with ICIs. Of note, this study specifically included patients with in-transit melanoma that is intrinsically immuno-resistant. The results showed that patients who had not received prior immunotherapy had significantly higher complete response rates (47% vs. 6%) compared to those who were pretreated. Additionally, overall survival and distant progression-free survival were significantly better in the ICI-naïve group. These findings suggest that the improved complete response observed after ILP may be at least partially immunologically mediated, consistent with experimental evidence [111]. However, there were no significant differences in overall response rates, stable disease, progressive disease, or local progression-free survival between the groups. Despite the reduced efficacy in patients pretreated with immunotherapy, ILP remains a valid second-line treatment option for local control in patients with advanced melanoma in-transit metastases who have failed immunotherapy, highlighting its role in comprehensive treatment strategies [104]. Similarly, a recent study by Rastrelli et al. examined the effects of combining ILP locoregional treatment with immunotherapy in a cohort of 187 ITM patients between 1989 and 2021. The study found that patients who received both ILP and immunotherapy experienced significant improvements in overall survival and disease-specific survival. Specifically, the overall survival at 36 months was 43% in the ILP-only group compared to 61% in the group that received both ILP and immunotherapy. Additionally, the disease-specific survival at 36 months was 43% in the ILP group and 64% in the ILP plus ICI group. These findings further underscore the potential benefits of integrating locoregional and systemic therapies for improved patient outcomes in advanced melanoma [105]. Eventually, another locoregional treatment—electrochemotherapy—has shown promising results when combined with the immune checkpoint inhibitor pembrolizumab (anti-PD-1) in several studies. In a comparative retrospective study, Campana et al. evaluated the efficacy of pembrolizumab plus ECT against both pembrolizumab alone and ECT alone in patients with stage IIIC–IV cutaneous melanoma metastases. Unlike other retrospective studies that typically evaluate the combination therapy and only one of the treatments alone, this study uniquely assessed the effects of each single treatment as well as the combination. The results indicated that the local response efficacy on cutaneous metastases was similar in the pembrolizumab plus ECT group and ECT alone group, both of which were significantly better than the pembrolizumab alone group. Furthermore, the combination of pembrolizumab with ECT also improved systemic progression-free survival, highlighting a synergistic effect between the two therapies [106].

While previous studies were retrospective and included patients with ITM incidentally, rather than being specifically designed to investigate the combination of locoregional and systemic therapies, there is now a growing focus on purpose-built studies. These new studies aim to document the effects of combining systemic therapies with locoregional treatments for in-transit metastases. Such increasing awareness underscores the benefits of a dual approach: systemic therapies provide comprehensive effects throughout the body, while locoregional treatments deliver higher drug dosages directly to the metastases. This combined strategy is emerging as a promising method for enhancing patient outcomes in ITM, e.g., the clinical trial NCT02557321 is investigating a combination treatment for patients with unresectable stage IIIB-IVM1c melanoma. This combination involves pembrolizumab, which is a systemic immune checkpoint inhibitor anti-PD-1, and PV-10, which is a locoregional treatment. PV-10 is a 10% solution of Rose Bengal used for intralesional chemoablation. It induces tumor cell lysis, thereby promoting a local anti-tumor response. Preliminary results from phase 1b showed promising effects, reaching ORR of 67% and 29% in ICI-naïve and ICI-refractory ITM patients, respectively [107,108]. As a matter of fact, combining locoregional and systemic treatments may potentially enhance therapeutic outcomes through synergistic effects. Indeed, earlier in vivo studies on the combination of PV-10 with other treatments have shown promise in advanced-stage diseases with substantial tumor burdens that are inaccessible to direct injection. Specifically, the immune stimulation resulting from PV-10-induced tumor ablation can complement and enhance the efficacy of nonspecific immunotherapies. Therefore, synergistic interaction can potentially lead to more effective tumor control and improved patient outcomes [112,113,114]. A key study, clinical trial NCT01323517, investigated the combination of systemic and locoregional therapies in patients with stage IIIB/IV melanoma. Phase II findings indicated that using Isolated Limb Infusion alongside ipilimumab (anti-CTLA-4) resulted in a one-year progression-free survival rate of 57%. In another trial, the NivoILP study, researchers assessed the safety and effectiveness of administering a single dose of the PD-1 inhibitor nivolumab in conjunction with Isolated Limb Perfusion for in-transit melanoma metastases. Twenty patients were randomized to receive either nivolumab or a placebo the day before ILP. The results showed a 75% complete response rate at three months in the nivolumab group, compared to 60% in the placebo group. Additionally, the one-year local progression-free rate was higher in the nivolumab group (86%) than in the placebo group (67%), with both groups showing a one-year overall survival rate of 100%. The study concluded that a single pre-ILP dose of nivolumab is safe and may enhance local control and PFS, suggesting further research in a Phase II trial to validate these outcomes and potentially establish a new treatment protocol for melanoma in-transit metastases [110].

Overall, current data indicate that combinatorial approaches may not be detrimental and could be beneficial for patients with in-transit melanoma metastases. However, further research is required to delineate which cohort of ITM patients would benefit most from receiving locoregional therapy as a first-line treatment, reserving immunotherapy for recurrences, or vice versa, or implementing both therapies concurrently. Additionally, the timing and sequencing of these therapies are critical factors that need to be investigated to optimize patient outcomes. Based on our clinical experience and the existing data, we propose a personalized treatment model that integrates systemic and locoregional therapies, tailored to the specific needs of each patient with ITM. This approach aims to maximize therapeutic efficacy and improve patient outcomes.

To provide a clear and structured guide in the management of ITM, we propose the following flow chart (Figure 2):

### Initial Evaluation and Staging

Effective management of ITM begins with an accurate and comprehensive assessment of the disease status and the patient’s condition. This includes comprehensive imaging such as Positron Emission Tomography-Computed Tomography, Computed Tomography, and Magnetic Resonance Imaging to determine the extent of the disease and identify any distant metastases. It is also crucial to assess the patient performance status using standardized scales such as the Eastern Cooperative Oncology Group (ECOG) Performance Status. Based on the results of the staging tests, we may find ourselves in one of the three possible scenarios. 

For patients with resectable ITM and no distant metastatic disease, surgical resection remains the first-line treatment. The goal of surgery is to achieve negative margins to reduce the risk of local recurrence. Subsequently, adjuvant systemic therapy should be considered, based on the patient’s overall condition and any comorbidities, to reduce the risk of recurrence and improve overall survival.For patients with ITM and metastatic disease, Primary Systemic Therapy with immune checkpoint inhibitors or targeted therapies is the recommended treatment.
If there is a good response to therapy on distant metastases and in-transit metastases, we recommend continuing with systemic treatment and possibly integrating it with a low-impact locoregional treatment (such as ECT) if a complete response is not achieved with systemic therapy;If there is a good response to therapy on distant metastases and a poor response on in-transit metastases, we recommend continuing with systemic treatment and integrating with the locoregional treatment that best suits the disease characteristics and patient features. For bulky disease (or high burden) of the limbs, we propose ILP for patients with good ECOG performance status, or ILI for patients with poor ECOG. ECT is advised for nodules smaller than 3 cm or for intralesional treatments;In the case of poor response to therapy on distant metastases, we advise changing systemic treatment, if possible.


If there is a subsequent systemic response, we revert to the previously considered cases.

In the case of systemic progression, we still propose considering low-impact locoregional treatments (such as ECT or laser) to manage symptoms due to in-transit metastases (pain, bleeding, and infections) and achieve an improvement in quality of life.

3.For patients with unresectable ITM and no distant metastatic disease, we propose integrating systemic therapy (if feasible) and the most appropriate locoregional treatment. For bulky disease (or high burden) of the limbs, we propose combining ILP and immunotherapy for patients with good ECOG or ILI and immunotherapy. For patients with small-sized in-transit metastases or a limited number of lesions, we propose the combination of ECT + immunotherapy or intralesional treatments + immunotherapy.

In case of local recurrence or poor local response, consider repeating locoregional treatment or another locoregional treatment, continuing immunotherapy unless distant metastases develop. In the latter case, the guidelines from point 2 can be followed.

## 8. Conclusions

In-transit metastases represent a significant challenge in the management of advanced cutaneous melanoma due to their clinical and biological complexity. The prognosis for patients with ITM is generally unfavorable, with 5- and 10-year survival rates varying. Recent innovations in systemic treatments, particularly the introduction of immune checkpoint inhibitors and targeted therapies, have greatly improved the outlook for these patients, increasing overall survival and progression-free survival rates. However, optimal management of ITM requires an integrated and personalized therapeutic approach, combining systemic and locoregional treatments. Locoregional therapies such as ILP, ILI, ECT, and intralesional therapies offer targeted solutions for controlling localized disease and improving patient quality of life. The combination of these therapies with systemic treatments can enhance therapeutic efficacy, as demonstrated by various clinical studies. The introduction of modern immunotherapies has not compromised the efficacy of locoregional therapies but has instead improved melanoma-specific survival. Current guidelines, including those from the NCCN, ESMO, and AIOM, recommend a multidisciplinary approach and the integration of systemic and locoregional therapies to optimize clinical outcomes. Surgery remains the first-line treatment for resectable ITM, followed by adjuvant therapies to reduce the risk of recurrence. For unresectable ITM or in the presence of metastatic disease, the combination of immunotherapy, targeted therapies, and locoregional treatments offers a promising strategy for improving survival and managing symptoms.

Here, we propose a personalized combined treatment model for patients with ITM, based on the integration of systemic and locoregional therapies. This approach is illustrated in our treatment flow chart (Figure 2), providing a clear and structured guide for managing ITM. Our proposal aims to further improve clinical outcomes by combining the local response capabilities of locoregional therapies with the benefits of systemic treatments in terms of overall survival and overall response.

In conclusion, the treatment of in-transit metastases from cutaneous melanoma requires an integrated and personalized therapeutic approach. Recent innovations in systemic and locoregional treatments have improved the outlook for these patients, but ongoing research is needed to optimize therapeutic combinations and further enhance clinical outcomes. The inclusion of patients in clinical trials remains essential to advance the understanding and management of this complex manifestation of melanoma.

## Figures and Tables

**Figure 1 cancers-16-03065-f001:**
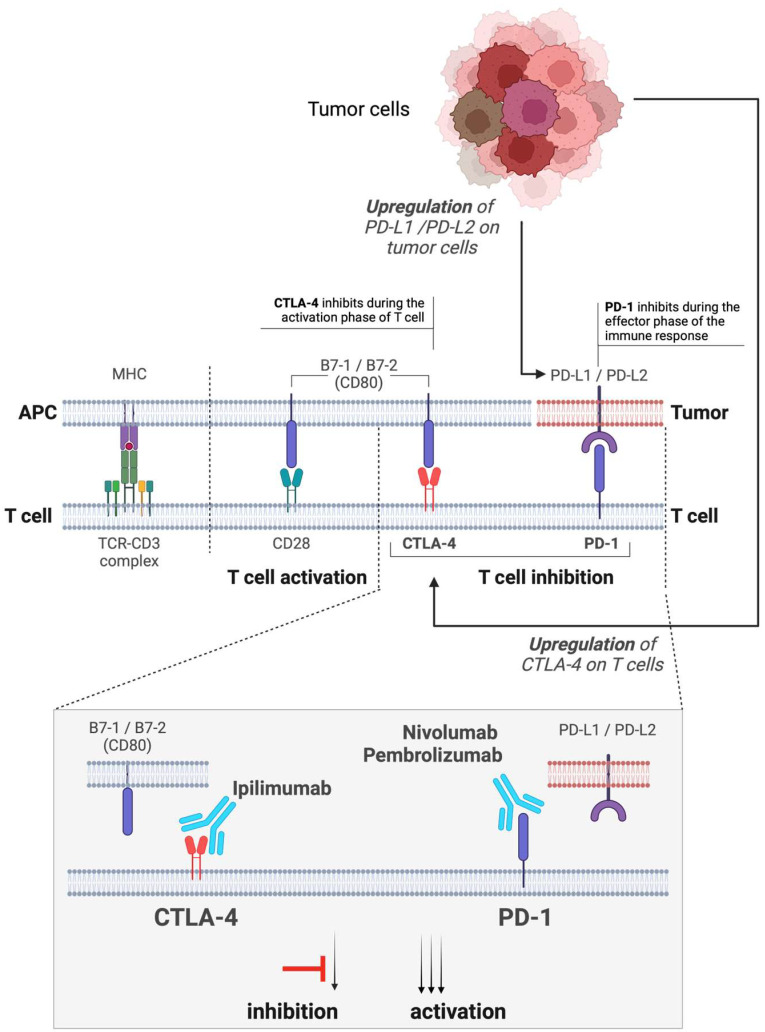
Mechanisms of T cell activation and inhibition and the role of immune checkpoint inhibitors in cancer therapy Upper Panel: tumor cells with upregulated expression of PD-L1 (Programmed Death-Ligand 1) and PD-L2 (Programmed Death-Ligand 2), which play key roles in suppressing the immune response. Antigen-Presenting Cell (APC): The MHC–TCR complex shows the interaction between the Major Histocompatibility Complex (MHC) on APCs and T cell receptors (TCR) on T cells, initiating T cell activation. B7-1/B7-2 (CD80) are co-stimulatory molecules on APCs that interact with CD28 on T cells, further promoting T cell activation. CTLA-4 and CD28 on T Cells: CTLA-4 is an inhibitory receptor on T cells that competes with CD28 for binding to B7-1/B7-2, leading to the inhibition of T cell activation. PD-1 on T Cells: Programmed cell death protein 1 (PD-1) on T cells interacts with PD-L1/PD-L2 on tumor cells, resulting in the inhibition of T cell activity. Lower Panel: Immune Checkpoint Inhibitors: Ipilimumab is a monoclonal antibody that blocks CTLA-4, preventing its interaction with B7-1/B7-2, thus promoting T cell activation. Nivolumab and Pembrolizumab are monoclonal antibodies that block PD-1, preventing its interaction with PD-L1/PD-L2 on tumor cells, thereby enhancing T-cell activity against tumors. Created with BioRender.com.

**Figure 2 cancers-16-03065-f002:**
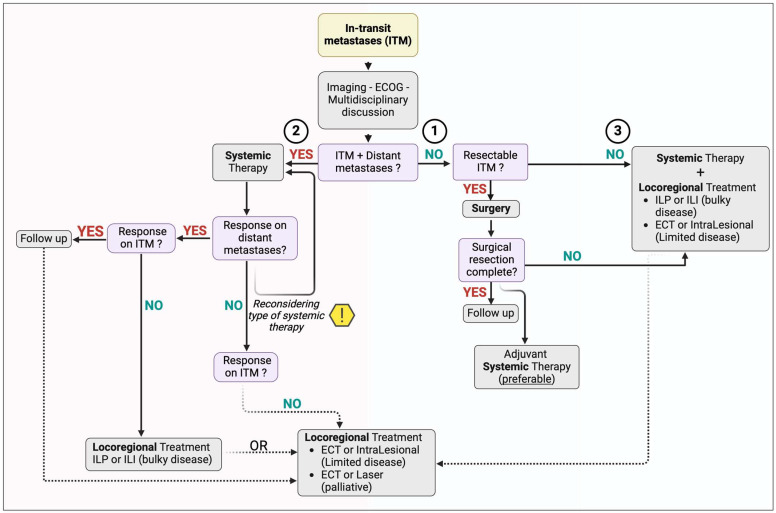
Flow chart guide for the management of in-transit melanoma, Isolated Limb Perfusion (ILP), Isolated Limb Infusion (ILI), Electrochemotherapy (ECT), and Eastern Cooperative Oncology Group (ECOG). The numbers correspond to the points listed in the “Initial Evaluation and Staging description. Created with *BioRender.com*.

**Table 1 cancers-16-03065-t001:** Efficacy of systemic therapies—alone or in combination with locoregional treatments—in ITM patients. ITM, in-transit metastases; ICI, immune checkpoint inhibitors; IT, immunotherapy; PFS, Progression-Free Survival; CR, Complete Response; OS, Overall Survival; ORR, Overall Response Rate; PD, Progressive Disease; PR, Partial Response, RFS, Relapse-Free Survival.

Reference	Other [%]	ORR [%]	CR [%]	OS [%]	PFS Rate [%]	Systemic (S)/Locoregional (L)	Treatment	No. Patients
Systemic treatments								
[29]		54	26 (2 years)	63 (2 years) (85% responders; 40% non-responders)	48 (1 year) 39 (2 years)	S	ICI (Anti-PD-1, anti-CTLA-4, Anti-PD-1/anti-CTLA-4)	58
[29]		58			30 (2 years)	S	Anti-PD-1	40
		38			50 (2 years)	S	Anti-CTLA-4	8
		40			80 (2 years)	S	Anti-PD-1/anti-CTLA-4	5
[100]	RFS, 54% (4 years)					S	BRAF/MEK inhibitors	870
Systemic and Locoregional treatments								
[28]	PD, 32	56	36		47 (1 year) 33 (2 years) 19 (5 years)	S/S + L	ICI (Anti-PD-1, anti-CTLA-4, Anti-PD-1/anti-CTLA-4)/ICI + Locoregional	287
	PD, 34	56	37			S/S + L	Anti-PD-1 (72/233 also locoregional)	233
[28]	PD, 35	43	30			S/S + L	Anti-CTLA-4 (12/23 also locoregional)	23
	PD, 23	68	35			S/S + L	Anti-PD-1/anti-CTLA-4 (14/31 also locoregional)	31
[104]		75	6	33 (2 years)		S + L	IT + ILP	18
			47	81 (2 years)		L	ILP	79
[105]	PD, 28	67	48	61 (3 years)		S + L	IT + ILP	88
	PD, 31		50	43 (3 years)		L	ILP	99
[106]	PD, 7 (local) PD, 38 (systemic)	78 (local) 25 (systemic)	49 (local) 11 (systemic)	88 (1 year) 70 (2 years)		S + L	Anti-PD-1-ECT	45
	PD, 27 (local) PD, 68 (systemic)	39 (local) 25 (systemic)	32 (local) 21 (systemic)	64 (1 year) 43 (2 years)		S	Anti-PD-1	44
	PD, 2	81 (local)	44 (local)			L	ECT	41
[107]		67				S + L	Anti-PD-1 + PV-10	21 (ICI-naïve)
[108]		29				S + L	Anti-PD-1 + PV-10	14 (ICI non-responders)
[109]			65 (3 months) (+ 24% PR)	78 (18 months)	57 (1 year)	S + L	Anti-CTLA-4 + ILI	18
[110]			75		86 (local)	S + L	Anti-PD-1 + ILP	10
			60		67 (local)	L	ILP	10

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
