# Peer review of "Therapeutic Treatment Options for In-Transit Metastases from Melanoma"

_cancers, 2024, doi:10.3390/cancers16173065_

Round 1

Reviewer 1 Report

Comments and Suggestions for Authors

General comments

Dear Authors, thank you for giving me an opportunity to review this manuscript.

The main content of research presented in the paper is the therapeutic treatment options for in-transit metastases from melanoma.

The presented study analyses recent multidisciplinary strategy to optimize the management of ITM in melanoma patients.

The main proposed paper is to investigate the role of combined modern immunotherapies and locoregional therapies in the management of ITM in melanoma highlighting the synergistic potential of combining these approaches. 

The topic is not unique but worthy of researching.

The article is well structured.  The progress of this paper is good compared with the current research results. The conclusions are tenable.

My observations are as follows:

INTRODUCTION

Line 87: “Microsatellite metastases are a variant of ITMs”. This statement is not true and must be corrected: a microsatellite metastasis is a focus of metastatic tumor in the dermis or subcutis that is adjacent to but discontinuous with the primary melanoma, and is identified during histopathologic assessment of the primary tumor excision. A satellite metastasis is a clinically evident cutaneous or subcutaneous metastasis that is within 2 cm of, but discontinuous with, the primary tumor. An in transit metastasis is a clinically evident cutaneous or subcutaneous metastasis greater than 2 cm from the primary melanoma, and typically situated between the primary melanoma and the regional lymph node basin. 

SYSTEMIC TREATMENT FOR IN-TRANSIT METASTASES

The authors report many concepts that are widely known. Advice to reduce content to the essential in order to make the reading more fluid.

LOCOREGIONAL TREATMENTS

For the treatment of melanoma metastases in the pelvic region, the two techniques used worldwide such as isolated limb perfusion and isolated limb infusion, cannot effectively target metastases. Pelvis relapse occurs in 15% of metastatic cutaneous melanoma and in this case a locoregional treatment option is hypoxic pelvic perfusions with hemofiltration. This treatment must be mentioned and quoted in the references (Guadagni S, Fiorentini G, Clementi M, Palumbo G, Chiominto A, Cappelli S, Masedu F, Valenti M. Melphalan hypoxic perfusion with hemofiltration for melanoma locoregional metastases in the pelvis. J Surg Res. 2017 Jul;215:114-124.   Guadagni S, Fiorentini G, Clementi M, Palumbo G, Palumbo P, Chiominto A, Baldoni S, Masedu F, Valenti M, Tommaso AD, Fabi B, Aliberti C, Sarti D, Guadagni V, Pellegrini C. Does Locoregional Chemotherapy Still Matter in the Treatment of Advanced Pelvic Melanoma? Int J Mol Sci. 2017 Nov 9;18(11):2382. Guadagni S, Zoras O, Fiorentini G, Masedu F, Lasithiotakis K, Sarti D, Farina AR, Mackay AR, Clementi M. A Prospective Study of Intraarterial Infusion Chemotherapy in Advanced Wild-Type BRAF Melanoma Patients. J Surg Res. 2021 Dec;268:737-747).

Strengths and weaknesses

  1. The title is attractive. 
  2. The abstract is informative. 
  3. The aim is clear.
  4. The KEYWORDS are good.
  5. The introduction provides sufficient background information for readers in the immediate field to understand the problem/hypotheses. 
  6. The review of the therapeutic options is good but the section is long and not easy to understand and should be simplified.
  7. The depth of the academic material is good.
  8. The readability should be improved.
  9. The results are good.
  10. All tables are clear enough to summarize the results for presentation to the readers
  11. All tables are well referred to in the text.
  12. The discussion of results from multiple angles is sufficient.
  13. The conclusion is good.
  14. The references are in order within the text.

Paper score

Ten-point System

Relevance

Originality

Significance

Technical soundness

Clarity of presentation / language

Enter a score between 0 to 10

8

8

9

9

7

Final Evaluation: Minor Revision

Length of the paper:

Should be extended

Perfect 

Should be shortened

ABSTRACT

X

PAPER

X

This paper should be published as:

Regular paper

X

Short paper 

Commentary Paper

Types of bias:

Follow up bias

Exclusion bias

Analysis bias

Reporting bias

X

Assembly bias

Measurement bias 

Detection bias 

Selection (susceptibility) bias 

Chronology bias

Channeling bias 

Response bias 

My final decision is acceptable after minor revision.

Comments on the Quality of English Language

Minor editing of English language required

Author Response

Please, see attached file including our response in a point-by-point manner.

Reviewer 2 Report

Comments and Suggestions for Authors

I am grateful to the editor for the opportunity to review this interesting paper. This review article addresses a highly relevant and timely topic within the oncological field of melanoma, with a particular focus on in-transit metastases (ITM), an area that is garnering significant attention in current literature due to its complex clinical implications. 

The manuscript is well-written and offers a comprehensive overview of both systemic and locoregional treatments, making it a valuable contribution to the ongoing discussion in this area of study.

I have a few suggestions to enhance the paper's fluidity and contextual clarity.

1. The paper provides a comprehensive analysis of available therapies for in-transit metastases (ITM) in melanoma. While the introduction is well-written, it would benefit from addressing the significance of ITM from both molecular and clinical perspectives. Specifically, I would suggest to explore why ITM is a critical subset of melanoma to study, differentiating these patients from those with other forms of advanced or metastatic melanoma.

Also, although the paper primarily focuses on clinical aspects, a brief discussion on the molecular underpinnings of ITM could be invaluable.  This could include insights into the unique molecular characteristics of ITM, such as specific genetic markers or signaling pathways that differentiate it from other metastatic melanoma processes.

Exploring how these molecular differences influence treatment approaches and impact patient outcomes would provide valuable context and justify the need for specialized management strategies for ITM.

2. Incorporating some key references could significantly strengthen the introduction and conclusion sections, adding important context to the suggestions made above.   

For instance, the paper by Lawless et al. provides critical insights into the clinicopathological characteristics that can predict recurrence and survival in patients with in-transit metastases. (Lawless AK, Coker DJ, Lo SN, Ahmed T, Scolyer RA, Ch'ng S, Nieweg OE, Shannon K, Spillane A, Stretch JR, Thompson JF, Saw RP. Clinicopathological characteristics predicting further recurrence and survival following resection of in-transit melanoma metastases. Ann Surg Oncol. 2022 Oct;29(11):7019-7028. doi: 10.1245/s10434-022-11997-0. Epub 2022 Jun 30.)

Similarly, the work by Wagstaff et al. on the molecular genetics and therapeutic resistance in melanoma could help to justify the need for specialized treatment strategies in ITM. (Wagstaff W, Mwamba RN, Grullon K, Armstrong M, Zhao P, Hendren-Santiago B, Qin KH, Li AJ, Hu DA, Youssef A, Reid RR, Luu HH, Shen L, He TC, Haydon RC. Melanoma: Molecular genetics, metastasis, targeted therapies, immunotherapies, and therapeutic resistance. Am J Pathol. 2001 Apr;158(4):1371-8. doi: 10.1016/S0002-9440(10)64088-6) 

The study by Jakub et al. (2022) highlights the association between tumor molecular factors and ITM, offering a bridge between molecular biology and clinical outcomes that could be referenced to emphasize the rationale behind combining systemic and locoregional therapies. (Jakub JW, Weaver AL, Meves A. Association of tumor molecular factors with in-transit metastasis in primary cutaneous melanoma. Int J Dermatol. 2022 Sep;61(9):1117-1123. doi: 10.1111/ijd.16141. Epub 2022 Mar 5.)

Lastly, the study by Nakayama et al., though relatively old, is important as it examines whether these metastases are clonal in origin, which would imply specific genetic alterations unique to this form of melanoma metastasis.(Nakayama T, Taback B, Turner R, Morton DL, Hoon DS. Molecular clonality of in-transit melanoma metastasis. Am J Pathol. 2001 Apr;158(4):1371-8. doi: 10.1016/S0002-9440(10)64088-6.)

In my opinion these references would help to substantiate the importance of a multidisciplinary approach in treating ITM. 

In conclusion, the paper presents a comprehensive and well-structured analysis of current therapeutic approaches for in-transit metastases in melanoma, making it a valuable contribution to the field.  These minor adjustments would enhance the manuscript, further strengthening its suitability for publication. 

Author Response

(The authors gave the same response as above.)

Reviewer 3 Report

Comments and Suggestions for Authors

The authors have generated a very well written review article on in-transit mets (ITM) of melanoma and its treatments. The flow is overall very well organized, relevant data cited. One part that may be included: the characteristics of patients with ITM, e.g., if there is any correlation with any kinds of mutation with ITM, demographic characteristics of these patients, the molecular features of ITM (its distinct features) etc. 

This is a very well written review overall. 

Author Response

(The authors gave the same response as above.)
